# A national cohort study of long-term opioid prescription and sociodemographic and health care-related risk factors

Cecilia Krüger [1] ✉, Johan Franck[2,3], Härje Widing[2], Jonas Hällgren [4], Mika Gissler [4,5,6,7] & Jeanette Westman[1,4,8]

## Abstract

**Background** Opioids are essential medicines for pain management; however, long-term use is associated with negative outcomes, including addiction. The aim of the study was to analyze the risk of long-term use after an initial opioid prescription and examine associated sociodemographic and health care-related risk factors.
**Methods** We identified a strictly defined, five-year opioid-naïve population of adults aged 18–64 years who received an initial opioid prescription between 2016 and 2020 in Swedish national registers. We modeled the association between individual characteristics and odds of long-term (>3 months) versus short-term (≤3 months) use, and odds of different durations of use (>3–6, >6–12, and >12 months) using logistic regression analyses.
**Results** Of 754,982 opioid-naïve individuals with an initial opioid prescription, 8.1% use opioids long-term. Individuals treated for a recent external injury have lower odds of long-term opioid use (e.g., >12 vs ≤3 months: OR 0.55, 95% CI 0.52–0.59), whereas those who initiated treatment in primary care have higher odds (e.g., >12 vs ≤3 months: OR 3.02, 95% CI 2.90–3.14). Individuals with a history of substance use disorders and greater use of psycholeptic drugs have higher odds of long-term use. Sociodemographic factors, including older age, lower education level, and not cohabiting are also associated with longer durations of use.
**Conclusions** Of opioid-naïve individuals, 8.1% develop long-term prescription opioid use, with higher odds among individuals with psychiatric history and whose opioid treatment initiated in primary care. Careful evaluation of patient health and regular follow-up are essential to reduce the risk of long-term opioid use.

## Plain language summary

Opioids are strong pain-relieving medicines, but long-term use can lead to serious health problems, including addiction. This study looked at how often Swedish adults continued using opioids after their first prescription, and what factors increased that risk. We found that 8 out of 100 people continued using prescription opioids for more than three months, and some for much longer periods of time. Those with mental health conditions, substance use disorders, or who received their first prescription in primary care were more likely to continue, while people treated for injuries such as broken bones had lower risk. These findings highlight the need to consider the risk of prolonged use before prescribing opioids, and to provide careful follow-up for patients who use them.

Opioid analgesic medications are essential in the management of pain, particularly after an acute injury, surgical intervention, or for palliative care[1]. However, prolonged use can lead to tolerance and physical dependence, and use of higher doses is associated with higher risk of negative outcomes, including hyperalgesia (i.e., heightened pain sensitivity), fatal and non-fatal overdoses, and addiction[1–4]. Misuse and nonmedical use of opioid prescriptions remains an international public health concern[5,6], and it is, therefore, important to continue to increase knowledge on the factors that

lead to long-term use. Improved understanding of risk factors for long-term use could help prevent opioid-related harms.

International[1,7] and Swedish[8] guidelines recommend restrictive prescribing with the lowest effective dose and duration. Despite limited evidence that opioids are effective in treating chronic non-cancer pain, many patients with chronic pain continue to receive long-term prescriptions[1,2,9,10]. Prevalence of long-term opioid use varies based on the population measured[9,10]. A systematic review showed long-term use in opioid-naïve

[1]Department of Neurobiology, Care sciences and Society, Karolinska Institutet, Huddinge, Sweden. [2]The Stockholm Centre for Dependency Disorders, Stockholm Health Care Services, Region Stockholm, Stockholm, Sweden. [3]Department of Clinical Neuroscience (CPF), Karolinska Institutet, Stockholm, Sweden. [4]Academic Primary Care Centre, Region Stockholm, Stockholm, Sweden. [5]Department of Molecular Medicine and Surgery, Karolinska Institutet, Stockholm, Sweden. [6]Department of Data and Analytics, Finnish Institute for Health and Welfare, Helsinki, Finland. [7]Department of Child Psychiatry, Turku University Hospital, Turku University, Turku, Finland. [8]Department of Health Care Sciences, Marie Cederschiöld University, Stockholm, Sweden. ✉e-mail: cecilia.kruger@ki.se

populations ranged from 0.2% to 35.8%[9]. The definition and measures used to assess long-term use also impact prevalence estimates. In epidemiological studies, long-term opioid use is often defined as use for longer than 3 months[9–12], which aligns with the definition of chronic pain[1,9,10] and the most frequent duration of dispensation in Scandinavian countries[13]. Attention on problems associated with overprescription of narcotics has increased globally as well as in Sweden[8], resulting in important changes in opioid prescribing patterns. Although there has been an overall decrease in the number of opioid prescriptions in the past decades[14], opioid-related harms remain a major public health concern.

The aim of this study was to analyze the risk of long-term use after an initial opioid prescription in a national cohort of five-year opioid-naïve adults and examine associated sociodemographic and health care-related risk factors. Leveraging comprehensive national data from Swedish registers, we were able to characterize and assess the risk of several important risk factors for prolonged prescription opioid use. Over 8% of first-time opioid users continue use for longer than three months following an initial prescription, with higher odds of long-term use among individuals who initiate treatment in primary care, have a history of psychiatric or substance use disorders, are older, have lower levels of education, or are not cohabiting. Results from this strictly defined cohort contribute to a better understanding of the factors influencing long-term opioid use, which could inform clinical practice and policies aimed at mitigating associated risks.

## Methods

### Ethics and inclusion

This register-based research project included local researchers employed in the healthcare system of Region Stockholm in the study design, analysis, and authorship processes. The study was approved by the Swedish Ethical Review Authority (Dnr. 2019-00516). In accordance with national regulations, informed consent was not required as the study utilized anonymized register data.

### Data sources

Following ethical approval, applications for access to individual-level data for research purposes were submitted to each register holder. These agencies conduct independent legal and ethical reviews in accordance with standard procedures and Swedish data protection laws before granting access to researchers affiliated with Swedish organizations.

The national statistics agency, Statistics Sweden, oversees access to the Total Population Register, which includes demographic data (age, sex, country of birth, and municipality of residence), and the Longitudinal Integrated Database for Health Insurance and Labor Market Studies (LISA), which includes information on cohabitation status and educational attainment. The National Board of Health and Welfare oversees access to three health-related registers we applied for: the National Patient Register (inpatient and outpatient health care-related characteristics, including diagnoses), the National Prescribed Drug Register (dispensed prescriptions), and the National Cause of Death Register (dates of death). Data from these registers were linked by the respective agencies using the unique personal identification number assigned to each Swedish resident. We received de-identified datasets in accordance with Swedish data protection regulations.

### Study population

We identified a nationwide cohort of adults aged 18 through 64 years who received an initial prescription of an opioid analgesic, classified under Anatomical Therapeutic Chemical (ATC) code N02A, during the observational period of January 1st, 2016 through December 31st, 2020. In order to assess health care history, individuals in the cohort were required to have been registered as living in Sweden for five years prior to receiving their first opioid prescription. The date of the initial opioid prescription following a 5-year opioid-naïve period was designated as the index date. In addition to excluding individuals who received an opioid prescription within five years, individuals with a prior diagnosis of opioid use disorder (OUD), defined as

an International Statistical Classification of Diseases and Related Health Problems 10th revision (ICD-10) diagnosis of code F11, and individuals who had received medication for OUD (ATC code N07BC) in the five years prior to the index date were excluded. Opioid use, measured from pharmacy-dispensed opioid prescriptions, was followed for one and a half years (545 days) after the index date. Individuals who died or emigrated during follow-up were excluded from the study. Individuals dispensed 4000 morphine milligram equivalents (MME) or more[15] during their initiation month (i.e., the first 30 days following index date), and those missing information on prescriber level (i.e., primary or specialized care) were also excluded (Fig. 1).

### Opioid use and duration of use

Opioid use was defined as having been dispensed any opioid analgesic (ATC code N02A), regardless of formulation (e.g., tablets, suppositories, transdermal patches). The following strong opioids were prescribed in Sweden during the study period (2016–2020), listed in order of frequency of dispensation: oxycodone, morphine, buprenorphine, oxycodone/naloxone, fentanyl, morphine/antispasmodics, tapentadol, ketobemidone, ketobemidone/antispasmodics, and hydromorphone[14]. Among weak opioids, the most commonly dispensed included codeine/paracetamol, tramadol, and codeine in combination with other non-opioid analgesics[14]. In addition, several other approved opioids were also eligible for inclusion in the study, even if they were rarely prescribed or later withdrawn by the Swedish Medical Products Agency. These include codeine/ibuprofen (formulations withdrawn 2017 and 2020), hydromorphone/antispasmodics, pethidine, pentazocine, tramadol/paracetamol (withdrawn 2021), tilidine, nalbuphine (withdrawn 2019), dextropropoxyphene, dextropropoxyphene/combinations excluding psycholeptics, and codeine/combinations excluding psycholeptics[14,16].

Dispensing dates were used to define the index date and duration of the opioid prescription. We used the refill gap method[17] to measure the duration of use, counting the number of days that elapsed between the index date and the date when the final opioid was dispensed during the study period. When the gap between dispensing dates exceeded 180 days, the individual was considered to have discontinued opioid use (see Fig. 2). The 180-day gap was chosen based on the 90-day maximum supply of medication in Sweden[13] plus a grace period of 90 days to allow for late prescription fills. Individuals who discontinued opioid use could not reenter the cohort.

Short-term use was defined as use for ≤3 months (≤90 days) and long-term use as use for >3 months (>90 days). Long-term use was also divided by duration into three periods: >3 to 6 months (91–180 days), >6 to 12 months (181–365 days), and >12 months (>365 days).

### Sociodemographic characteristics

The selection of characteristics was guided by published literature on risk factors for long-term use, which often overlap with known risk factors for opioid misuse and addiction[9,10]. Sociodemographic characteristics at the time of initial prescription included age (18–34, 35–49, 50–64 years), legal sex as in the Total Population Register (male, female), cohabitation status (cohabiting, not cohabiting), level of education (primary, secondary, tertiary), country of birth (Sweden, other European, other), and size of municipality of residence (smaller towns/rural areas, medium-sized towns, large cities). These variables were selected based on their relevance to healthcare access and prescribing patterns, and may also serve as proxies for broader sociodemographic factors. For detailed definitions of variables and data sources, see Supplementary Data 1.

### Health care-related characteristics

We examined health care-related characteristics preceding opioid prescription, including cancer diagnosis (yes/no) one year prior to initial prescription and recent external injury (yes/no), which was diagnosed in specialized care in the 3 months prior to index date and was included as a proxy for recent acute injury (Supplementary Data 1). Several variables related to psychiatric history were assessed, including diagnosis of substance

use disorders (yes/no, excluding OUD) and other mental health disorders (yes/no) in the five years prior to index date. Number of psycholeptic drug classes prescribed in the five years prior to index date was counted as a proxy for treatment of common psychiatric disorders (0–3 classes: benzodiazepines, antidepressants, antipsychotics). Prescription of each of the drug classes (yes/no) was also examined separately. The analyses included information on prescriber level (primary or specialized care) of the initial opioid prescription.

## Opioid characteristics

Characteristics of the opioid(s) prescribed on the index date included substance name, duration of action (long-acting/short-acting), and opioid strength (strong/weak). A conversion factor was applied to calculate the total amount of opioids prescribed on the index date to oral morphine milligram equivalents (MMEs) (see Supplementary Data 2). For the initiation month, total amount of opioids in MMEs and number of prescription fills was assessed.

## Statistics and reproducibility

Descriptive statistics on sociodemographic and health care-related characteristics were presented as percentages or means with standard deviations (SDs). Differences between groups were assessed using ANOVA for continuous variables and the chi-square test for categorical variables.

We used binary and multinomial logistic regression models to assess the association between risk factors and duration of opioid use, presented as odds ratios (ORs) with 95% confidence intervals (CIs). Sociodemographic variables included in the models were age, sex, cohabitation status, level of education, country of birth, and size of municipality of residence. Health care-related variables included in the models were history of cancer, recent external injury, substance use disorders, and number of common psychiatric disorders, as well as prescriber level. The binary logistic regression model evaluated the odds of long-term (>3 months) opioid versus short-term (≤3 months) use. We also conducted a multinomial logistic regression model that compared different durations of use (>3–6, >6–12, and >12 months) with short-term (≤3 months) use.

Given the wide range of definitions of long-term use in the literature, we conducted two sensitivity analyses using progressively stricter refill gap criteria. In the first analysis, we shortened the grace period for late refills by 60 days, resulting in a 120-day refill gap. In the second, more stringent analysis, we reduced the refill gap to 90 days where, in order to be classified as having long-term use, an individual was required to have at least 2 prescriptions dispensed within days 0–90 and at least one more days 91–180.

All analyses were performed with SAS statistical software version 9.4. The analytical code used to generate the results presented in this study is publicly available[18].

## Reporting summary

Further information on research design is available in the Nature Portfolio Reporting Summary linked to this article.

## Results

### Cohort characteristics

A total of 754,982 individuals initiated opioid therapy between 2016 and 2020 after the minimum 5-year opioid washout period. Of these, 694,123 individuals (91.9%) used opioids short-term (≤3 months) and 60,859 (8.1%) fulfilled the criteria for long-term use (>3 months). A decreasing proportion of the study population were dispensed opioids over longer durations: 4.6% used opioids for >3 to 6 months, 1.9% for >6 to 12 months, and 1.5% for >12 months (Fig. 1).

The study cohort was relatively evenly distributed in age (18–34 years, 29.0%; 35–49 years, 31.7%; 50–64 years, 39.2%; $p < 0.0001$) and sex (female, 52.3%; $p < 0.0001$). Individuals with long-term use had a lower level of education ($p < 0.0001$) and a lower proportion of cohabitation (p < 0.0001) at inclusion compared to those with short-term use (Table 1). As the duration of opioid use increased, the proportion who lived in smaller- and

**Fig. 1 | Flowchart of study population and exclusion criteria.**

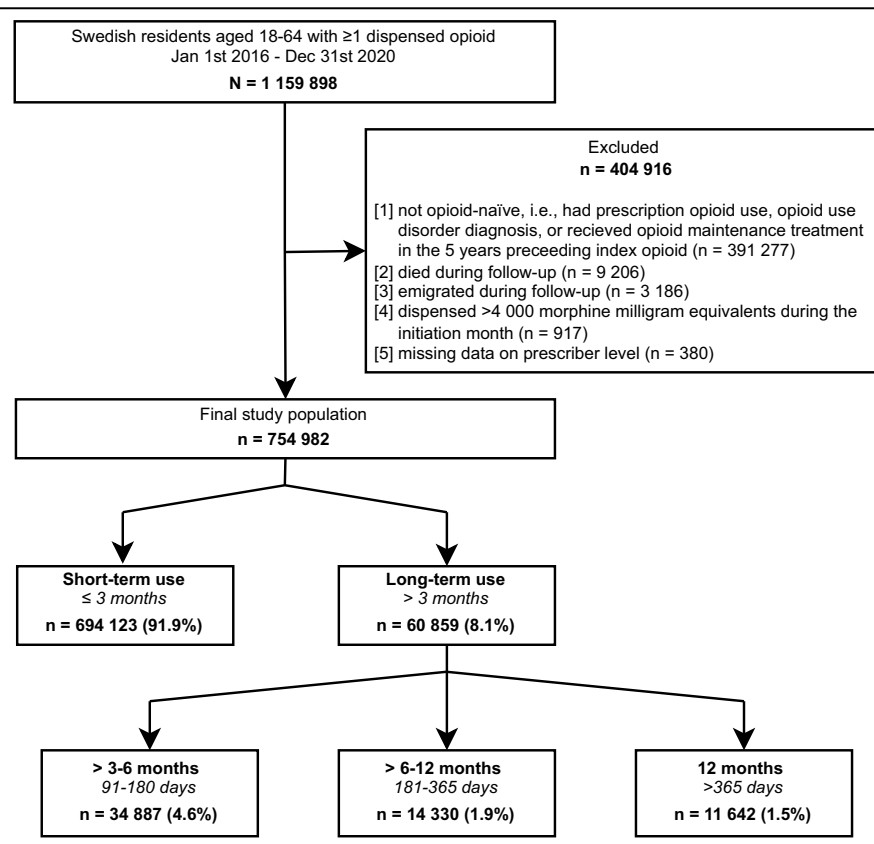

medium-sized towns increased and the proportion who lived in large cities decreased ($p < 0.0001$).

Relatively few individuals in the cohort had received a cancer diagnosis in the year prior to receiving their first opioid (3.2%), but a higher proportion of individuals with long-term use had cancer (>3–6 months, 5.3%; >6–12 months, 5.6%; >12 months, 7.2%; $p < 0.0001$) compared with short-term users (≤3 months, 3.0%). Approximately 20% of the cohort were diagnosed with an external injury in the 3 months preceding opioid initiation. Recent external injury was more common among individuals with short-term use, and the proportion decreased with longer durations of use ($p < 0.0001$). Individuals with long-term use had more mental health disorders and were treated with psycholeptic medications to a greater extent during the five years before the index date. Proportions increased with duration of opioid use, with significant differences between groups ($p < 0.0001$). For example, compared with individuals who used opioids for ≤3 months, more than twice as many who used opioids for >12 months had previously received a benzodiazepine prescription (15.5% vs. 38.9%; $p < 0.0001$). Short-term users of opioids more frequently received their initial prescription from specialized care (69.2%), and the proportion of users who received their index prescription from primary care increased with duration of use (>3–6 months, 41.8%; >6–12 months, 50.9%; >12 months, 60.6%; $p < 0.0001$).

## Opioid characteristics
The proportion of individuals with long-term use differed by opioid substance ($p < 0.0001$). Over three-quarters of the total study population were initially prescribed either codeine combination products (41.4%) or oxycodone (35.9%). An increasing proportion of individuals initially prescribed buprenorphine, fentanyl, tapentadol, and tramadol had longer durations use, whereas the proportion of individuals who initiated on morphine or morphine combination products, oxycodone, and oxycodone/naloxone decreased with longer durations of use (Table 2). The proportion of individuals receiving other substances was relatively constant. Initiation with strong opioids was only slightly more common (51.9%) than weak opioids in the overall cohort. An increasing proportion of individuals in the study population who initiated on weak opioids had long-term opioid use, with the highest proportion seen among those with the longest use (57.0% at

>12 months; $p < 0.0001$). A majority of the cohort (78.5%) initiated on opioids with short-acting mechanisms of action. However, the proportion of individuals who initiated on long-acting opioids increased with duration of use among long-term users ($p < 0.0001$). Individuals with short-term use had lower mean MME at index (218.9, SD 235.8) compared to long-term users, who had increasingly higher index doses of opioids ($p < 0.0001$).

Most of the study population filled their prescription only once during the initiation month, but as the duration of use increased, individuals had more prescription refills ($p < 0.0001$) and higher doses ($p < 0.0001$) in that first month. Those who used opioids for >12 months received 2.1 fills on average (SD 1.4) and a mean total of 630.5 MMEs (SD 715.3) during their initial 30 days of treatment.

## Regression model of long- versus short-term use
**Sociodemographic characteristics.** Older age (e.g., 50–64 vs. 18–34 years: OR 1.48, 95% CI 1.45–1.51) and lower levels of education (e.g., primary vs. tertiary: OR 1.49, 95% CI 1.45–1.53) were associated with higher odds of long-term than short-term opioid use (Fig. 3). Sex was not statistically significant (female: OR 0.99, 95% CI 0.97–1.00). Cohabiting was associated with lower odds of long-term use (OR 0.90, 95% CI 0.88–0.91), whereas being born outside Sweden or Europe was associated with slightly higher odds of long-term use (e.g., other European vs. Sweden: OR 1.04, 95% CI 1.01–1.07; other vs. Sweden: OR 1.08, 95% CI 1.05–1.11). Individuals living in smaller towns/rural areas or medium-sized towns had higher odds of long-term use than those living in large cities.

**Health care-related characteristics.** Individuals who had received a cancer diagnosis in the year preceding their first opioid prescription had higher odds of long-term use (OR 2.08, 95% CI 2.0–2.16). Inversely, individuals with a recent external injury had lower odds of continuing with opioid treatment (OR 0.74, 95% CI 0.72–0.76). Having a history of a substance use disorder (excluding OUD) resulted in higher odds of long-term use than in individuals without such disorders (OR 1.46, 95% CI 1.40–1.52). The higher the number of treatments for common psychiatric disorders in the five years before the index date, the higher the odds of long-term use (e.g., 1 vs. 0: OR 1.57, 95% CI 1.53–1.60). Those who

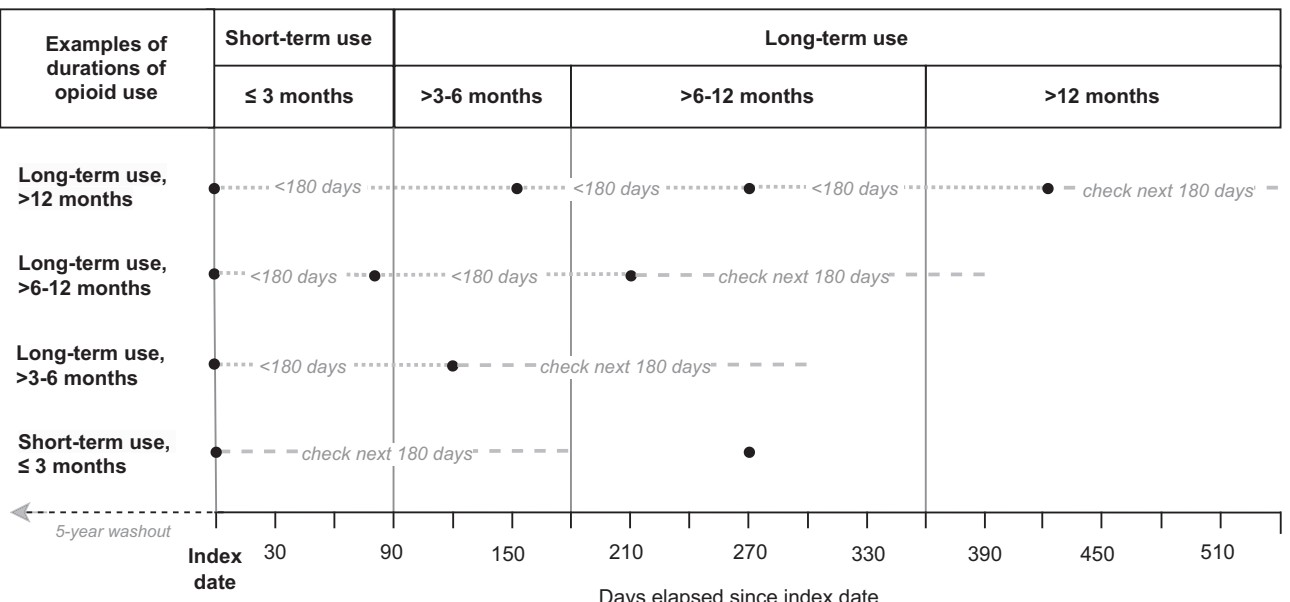

**Fig. 2 | Examples of opioid dispensing and definitions of opioid use using the gap refill method.** Each point in the figure indicates a dispensed opioid prescription. Duration of use was assessed by checking if another opioid was dispensed to the same individual within 180 days of the last dispensed opioid (represented by a dashed line). When the refill gap exceeded 180 days, individuals were considered to have discontinued use and could not reenter the cohort.

**Table 1 | Sociodemographic and health care-related characteristics by duration of opioid use**

| Characteristics | Short-term use | Duration of long-term use | | | p value |
|---|---|---|---|---|---|
| | ≤3 months n = 694,123 | >3–6 months n = 34,887 | >6–12 months n = 14,330 | >12 months n = 11,642 | |
| **Sociodemographic characteristics** | | | | | |
| Age group, n (%) | | | | | <0.0001 |
| 18–34 | 205,829 (29.7) | 8313 (23.8) | 3005 (21.0) | 2107 (18.1) | |
| 35–49 | 221,271 (31.9) | 10,721 (30.7) | 4281 (29.9) | 3359 (28.9) | |
| 50–64 | 267,023 (38.5) | 15,853 (45.4) | 7044 (49.2) | 6176 (53.0) | |
| Female sex, n (%) | 331,714 (47.8) | 16,141 (46.3) | 6699 (46.7) | 5535 (47.5) | <0.0001 |
| Cohabiting, n (%) | 401,835 (57.9) | 19,612 (56.2) | 7768 (54.2) | 5680 (48.8) | <0.0001 |
| Level of education, n (%) | | | | | <0.0001 |
| Primary | 95,057 (13.7) | 5731 (16.4) | 2668 (18.6) | 2588 (22.2) | |
| Secondary | 337,881 (48.7) | 17,792 (51) | 7544 (52.6) | 6357 (54.6) | |
| Tertiary | 261,185 (37.6) | 11,364 (32.6) | 4118 (28.7) | 2697 (23.2) | |
| Country of birth, n (%) | | | | | <0.0001 |
| Sweden | 559,907 (80.7) | 27,444 (78.7) | 11,286 (78.8) | 9440 (81.1) | |
| Other European | 59320 (8.5) | 3327 (9.5) | 1373 (9.6) | 1113 (9.6) | |
| Other | 74 896 (10.8) | 4116 (11.8) | 1671 (11.7) | 1089 (9.4) | |
| Size of municipality of residence, n (%) | | | | | <0.0001 |
| Smaller towns/rural areas | 145,249 (20.9) | 7648 (21.9) | 3370 (23.5) | 3169 (27.2) | |
| Medium-sized towns | 264,118 (38.1) | 13,489 (38.7) | 6019 (42) | 5207 (44.7) | |
| Large cities | 284,756 (41) | 13,750 (39.4) | 4941 (34.5) | 3266 (28.1) | |
| **Health care-related characteristics[a]** | | | | | |
| Cancer, n (%) | 20,550 (3) | 1857 (5.3) | 806 (5.6) | 835 (7.2) | <0.0001 |
| Recent external injury, n (%) | 139 711 (20.1) | 5288 (15.2) | 1915 (13.4) | 1119 (9.6) | <0.0001 |
| Substance use disorders, n (%) | 21,468 (3.1) | 1623 (4.7) | 884 (6.2) | 1002 (8.6) | <0.0001 |
| Other mental health disorders, n (%) | 75,747 (10.9) | 4965 (14.2) | 2374 (16.6) | 2506 (21.5) | <0.0001 |
| Number of psycholeptic drug classes[b], n (%) | | | | | <0.0001 |
| 0 | 501,345 (72.2) | 21,762 (62.4) | 7978 (55.7) | 5083 (43.7) | |
| 1 | 125,345 (18.1) | 7934 (22.7) | 3626 (25.3) | 3345 (28.7) | |
| 2 | 56,455 (8.1) | 4245 (12.2) | 2200 (15.4) | 2475 (21.3) | |
| 3 | 10,978 (1.6) | 946 (2.7) | 526 (3.7) | 739 (6.3) | |
| Psycholeptic prescription medications | | | | | |
| Benzodiazepines, n (%) | 107,686 (15.5) | 7925 (22.7) | 4039 (28.2) | 4526 (38.9) | <0.0001 |
| Antidepressants, n (%) | 145,114 (20.9) | 9910 (28.4) | 4793 (33.4) | 4954 (42.6) | <0.0001 |
| Antipsychotics, n (%) | 183,89 (2.6) | 1427 (4.1) | 772 (5.4) | 1032 (8.9) | <0.0001 |
| Prescriber level, primary care, n (%) | 213,556 (30.8) | 14,594 (41.8) | 7291 (50.9) | 7053 (60.6) | <0.0001 |

n = 754 982 opioid-naïve individuals included in the analysis.

[a]Based on diagnoses in period prior to initial prescription: 1-year history of cancer, 3-month history of an external injury, and 5-year history of substance use disorders and number of psycholeptic medications (see Supplementary Data 1).

[b]Individual-level, 5-year history of prescription psycholeptic drug use is a proxy for number of common psychiatric disorders.

received their initial prescription from primary care had significantly higher odds of long-term use compared to those who received their initial prescription in specialized care (OR 1.88, 95% CI 1.85–1.92).

**Regression model for duration of long-term opioid use**
**Sociodemographic characteristics**. Older age and lower educational levels were associated with higher odds of every category of long-term use when compared with short-term use (Table 3). The longer the use, the greater the association with older age and with lower educational levels. The results on odds of long-term opioid use by sex were mixed. Females had slightly higher but not statistically significant odds of using opioids for shorter durations of prolonged use, which decreased thereafter, resulting in a statistically significant lower risk of using opioids for >12 months (OR 0.90, 95% CI 0.86–0.93). Cohabitation was associated with lower odds of long-term use, and the association strengthened as the duration of opioid use increased.

The analysis showed that individuals born in a European country other than Sweden had 6% higher odds of using opioids for >3 to 6 months (OR 1.06, 95% CI 1.02–1.10), but the difference was statistically insignificant for longer use. When compared with short-term users, individuals born outside Europe had higher odds of using opioids for >3 to 6 months (OR 1.11, 95% CI 1.08–1.15) and >6 to 12 months (OR 1.12, 95% CI 1.06–1.18) but lower odds of using opioids for >12 months (OR 0.91, 95% CI 0.85–0.97). Compared with those living in large cities, individuals who lived in medium-

**Table 2 | Characteristics of index opioid prescription and use during the initiation month by duration of opioid use**

| Characteristic | Short-term use | Duration of long-term use | | | |
|---|---|---|---|---|---|
| | ≤ 3 months *n* = 694,123 | >3–6 months *n* = 34,887 | >6–12 months *n* = 14,330 | >12 months *n* = 11,642 | *p* value |
| Characteristics of initial opioid prescription | | | | | |
| Substance name, *n* (%) | | | | | <0.0001 |
| Buprenorphine | 1629 (0.2) | 252 (0.7) | 205 (1.4) | 410 (3.5) | |
| Fentanyl | 138 (0.0) | 27 (0.1) | 18 (0.1) | 52 (0.4) | |
| Hydromorphone | 1 (0.0) | 1 (0.0) | — | — | |
| Ketobemidone and combination products[a] | 3148 (0.5) | 167 (0.5) | 55 (0.4) | 41 (0.4) | |
| Codeine combination products[a,b] | 286,361 (41.3) | 15,131 (43.4) | 6161 (43.0) | 4836 (41.5) | |
| Morphine and combination products[a] | 55,440 (8.0) | 2947 (8.4) | 1011 (7.1) | 629 (5.4) | |
| Oxycodone | 252,529 (36.4) | 10,830 (31) | 4238 (29.6) | 3218 (27.6) | |
| Oxycodone/naloxone | 41,683 (6.0) | 1890 (5.4) | 668 (4.7) | 406 (3.5) | |
| Tapentadol | 1023 (0.1) | 70 (0.2) | 61 (0.4) | 92 (0.8) | |
| Tramadol | 44,070 (6.3) | 3116 (8.9) | 1688 (11.8) | 1789 (15.4) | |
| ≥2 opioid classes dispensed | 8101 (1.2) | 456 (1.3) | 225 (1.6) | 169 (1.5) | |
| Opioid strength[c], strong, *n* (%) | 363,400 (52.4) | 16,621 (47.6) | 6459 (45.1) | 5001 (43.0) | <0.0001 |
| Duration of action, long-acting, *n* (%) | 148 924 (21.5) | 7 354 (21.1) | 3 240 (22.6) | 2 836 (24.4) | <0.0001 |
| Morphine milligram equivalent (MME)[d] index[d] | | | | | |
| Mean (SD) | 218.9 (235.8) | 252.3 (296.6) | 285.4 (329.2) | 325.6 (402.1) | <0.0001 |
| Quartiles, Q1, Q3 | 105.0, 250.0 | 105.0, 300.0 | 105.0, 315.0 | 120.0, 375.0 | |
| Characteristics of initiation month (30 days after initial opioid) | | | | | |
| MME initiation month (30 days)[d] | | | | | |
| Mean (SD) | 266.6 (314.4) | 387.9 (486.4) | 508.9 (587.9) | 630.5 (715.3) | <0.0001 |
| Quartiles, Q1, Q3 | 105.0, 315.0 | 105.0, 420.0 | 150.0, 630.0 | 150.0, 840.0 | |
| Number of prescription fills | | | | | |
| Mean (SD) | 1.4 (0.8) | 1.7 (1.2) | 2.0 (1.4) | 2.1 (1.4) | <0.0001 |

*n* = 754 982 opioid-naïve individuals included in the analysis for all variables except MME index and MME initiation month (see footnote d).

*MME* morphine milligram equivalent, *Q* quartile, *SD* standard deviation.

[a]Includes single-ingredient and combination products with the same active substance.

[b]Substances in codeine combination products include paracetamol, acetylsalicylic acid, ibuprofen.

[c]Strong opioids include oxycodone, morphine, hydromorphone, fentanyl, transdermal and sublingual buprenorphine, ketobemidone, and tapentadol. Weak opioids include codeine, codeine combination products, and tramadol.

[d]Dispensations of drugs with missing dose-specific information (*n* = 40,622) marked in Supplementary Data 2 were excluded from the calculation of MMEs, resulting in a lower number of observations for the variables MME index and initiation month.

sized towns or smaller towns/rural areas had higher odds of using opioids for >6 to 12 months and >12 months, but the results were on the borderline of statistical significance for use for >3 to 6 months.

**Health care-related characteristics.** History of cancer in the year before initiating opioids was associated with increasing odds of all durations of long-term use, with odds ratios spanning from 1.86 (95% CI 1.77–1.95) for >3 to 6 months to 2.94 (2.73–3.17) for >12 months of use (Table 3). The opposite was observed for recent external injuries, as individuals with an injury diagnosis in the 3 months before initiation had lower odds of receiving opioids for longer durations. The more treatments for common psychiatric disorders individuals had received in the five years before their initial opioid prescription, the higher the odds that they would use opioids longer. In individuals with a history of treatment with all three classes of psycholeptic drugs, the odds ratio ranged from 1.77 (95% CI 1.65–1.90) for >3 to 6 months of use to 4.97 (95% CI 4.57–5.41) for >12 months of use. The same pattern was observed in individuals diagnosed with any substance use disorder, who had 30% higher odds of using opioids for >3 to 6 months (OR 1.30, 95% CI 1.23–1.37) to 75% higher odds of using opioids for >12 months (OR 1.75, 95% CI 1.63–1.88). Those first prescribed opioids in primary care had higher odds of continuing to use the drugs than individuals whose first

prescription came from specialized care, spanning from an odds ratio of 1.54 for >3 to 6 months (95% CI 1.50–1.57) to 3.02 for >12 months (95% CI 2.90–3.14).

**Sensitivity analyses**

Two sensitivity analyses were conducted using 120-day and 90-day refill gaps between dispensing dates. The prevalence of long-term opioid use was 5.1% with the 120-day gap and further decreased to 3.0% with the 90-day gap.

In the logistic regression models comparing short- and long-term use (Supplementary Figs. 1–2), the magnitude of the associations increased with stricter criteria for defining long-term use. For example, the odds ratio for female sex decreased, indicating a statistically significantly lower risk of use for >3 months in the 120-day model, and even lower risk in the 90-day refill gap model. Only one association changed direction: as estimates for country of birth decreased, being born outside Sweden shifted from a borderline statistically significant higher risk of long-term use in the primary analysis, to non-significant in the 120-day sensitivity analysis, and to a small but significant lower risk in the 90-day analysis.

The multinomial regression models (Supplementary Tables 1-2) provide more detailed insights across different durations of use and showed stronger associations for all health care-related characteristics under stricter

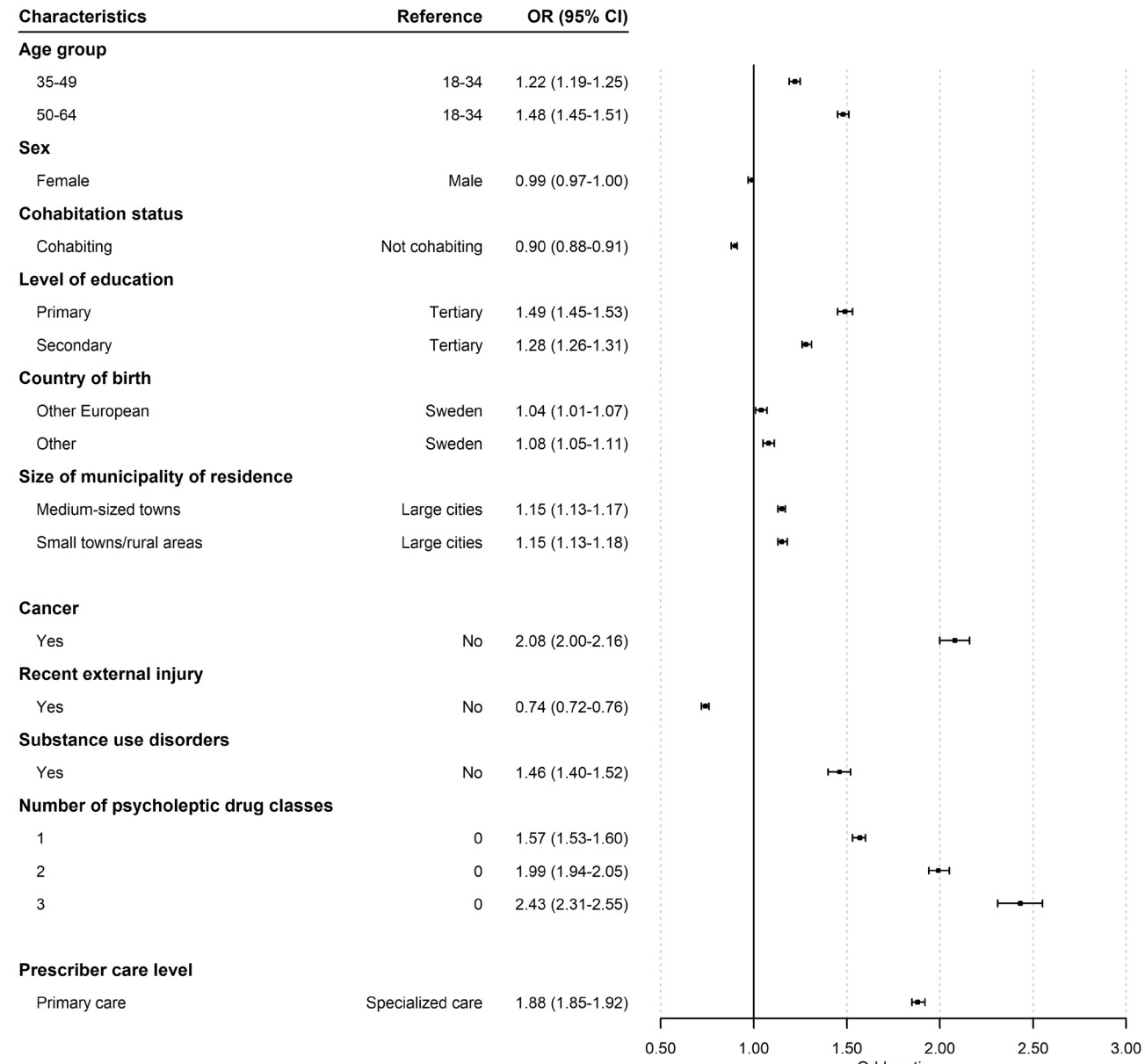

**Fig. 3 | Odds ratios (ORs) and 95% confidence intervals (CIs) for long-term opioid use (>3 months) (reference: short-term use defined as ≤3 months).** Abbreviations: CI confidence interval, OR odds ratio. *n* = 754 982 opioid-naïve individuals included in the analysis. Health care-related characteristics are based on diagnoses in period prior to initial prescription: 1-year history of cancer, 3-month history of an external injury, and 5-year history of substance use disorders and number of psycholeptic medications. History of prescription psycholeptic drug use is a proxy for number of common psychiatric disorders.

refill gap criteria in the sensitivity analyses. For female sex, the risk was significantly lower for >6 to 12 month use in the 120-day model and protective across all durations in the 90-day model. Individuals in smaller towns/rural areas, and medium-sized towns had consistently higher odds, becoming statistically significant across all durations. Country of birth was the only characteristic with a directional shift: being born in another European country was no longer a significant risk factor for >3-to 6-month opioid use in the 120-day model and was associated with significantly lower odds for >6 to 12 and >12 month use in the 90-day model. Similarly, for individuals born outside Europe, significance declined with stricter criteria, becoming non-significant for >6 to 12 month use in the 120-day model and >3 to 6 months in the 90-day model.

## Discussion

This study investigated long-term prescription opioid use in a strictly defined five-year opioid-naive, nationwide cohort of adults <65 years between 2016 and 2020. Approximately 8% of first-time opioid users in the population progressed to long-term use following their initial prescription. Having received the initial opioid prescription in primary care was associated with higher risk of long-term use (>3 months) and longer durations of use (>3–6, >6–12, and >12 months), whereas individuals treated for a recent external injury had lower risk. Individuals with a history of psychiatric or substance use disorders were also at higher risk of progressing to long-term opioid use of all durations, as were persons with older age, lower levels of education, and those not cohabiting.

The findings of this study highlight the role of primary care, from which one third of initial opioid prescriptions in this cohort originated, in long-term opioid prescribing. Individuals whose first prescription came from a primary care provider had higher odds of long-term use compared to individuals with an initial prescription from specialized care. Moreover, receiving the index prescription in primary care was associated with longer durations of use, with approximately three-fold higher odds of using opioids

**Table 3 | Odds ratios (ORs) and 95% confidence intervals (CIs) of the duration of long-term opioid use (reference: short-term use defined as ≤3 months)**

| Characteristics | Reference | >3–6 months OR (95% CI) | >6–12 months OR (95% CI) | >12 months OR (95% CI) |
|---|---|---|---|---|
| **Sociodemographic characteristics** | | | | |
| Age group | | | | |
| 35–49 | 18–34 | 1.15 (1.12–1.19) | 1.26 (1.20–1.32) | 1.41 (1.33–1.49) |
| 50–64 | 18–34 | 1.35 (1.31–1.39) | 1.59 (1.52–1.66) | 1.85 (1.76–1.95) |
| Sex | | | | |
| Female | Male | 1.02 (1.00–1.04) | 0.98 (0.95–1.01) | 0.90 (0.86–0.93) |
| Cohabitation status | | | | |
| Cohabitating | Not cohabiting | 0.94 (0.92–0.97) | 0.89 (0.86–0.93) | 0.77 (0.74–0.80) |
| Level of education | | | | |
| Primary | Tertiary | 1.31 (1.27–1.35) | 1.57 (1.49–1.65) | 2.12 (2.01–2.25) |
| Secondary | Tertiary | 1.19 (1.16–1.22) | 1.33 (1.28–1.38) | 1.59 (1.51–1.66) |
| Country of birth | | | | |
| Other European | Sweden | 1.06 (1.02–1.10) | 1.04 (0.98–1.10) | 0.98 (0.92–1.04) |
| Other | Sweden | 1.11 (1.08–1.15) | 1.12 (1.06–1.18) | 0.91 (0.85–0.97) |
| Size of municipality of residence | | | | |
| Medium-sized towns | Large cities | 1.02 (1.00–1.05) | 1.24 (1.19–1.29) | 1.55 (1.48–1.62) |
| Smaller towns/rural areas | Large cities | 1.03 (1.00–1.06) | 1.21 (1.15–1.26) | 1.59 (1.51–1.68) |
| **Health care-related characteristics[a]** | | | | |
| Cancer | | | | |
| Yes | No | 1.86 (1.77–1.95) | 2.09 (1.94–2.25) | 2.94 (2.73–3.17) |
| Recent external injury | | | | |
| Yes | No | 0.80 (0.77–0.82) | 0.74 (0.71–0.78) | 0.55 (0.52–0.59) |
| Substance use disorders | | | | |
| Yes | No | 1.30 (1.23–1.37) | 1.56 (1.45–1.68) | 1.75 (1.63–1.88) |
| Number of psycholeptic drug classes[b] | | | | |
| 1 | 0 | 1.37 (1.33–1.40) | 1.65 (1.58–1.72) | 2.30 (2.19–2.40) |
| 2 | 0 | 1.58 (1.52–1.63) | 2.12 (2.02–2.23) | 3.52 (3.35–3.71) |
| 3 | 0 | 1.77 (1.65–1.90) | 2.48 (2.25–2.72) | 4.97 (4.57–5.41) |
| Prescriber level | | | | |
| Primary care | Specialized care | 1.54 (1.50–1.57) | 2.14 (2.06–2.21) | 3.02 (2.90–3.14) |

n = 754 982 opioid-naïve individuals included in the analysis.

*CI* confidence interval, *OR* odds ratio.

[a]Based on diagnoses in period prior to initial prescription: 1-year history of cancer, 3-month history of an external injury, and 5-year history of substance use disorders and number of psycholeptic medications (Supplementary Data 1).

[b]Individual-level, 5-year history of prescription psycholeptic drug use is a proxy for number of common psychiatric disorders.

for over a year. The reason for opioid prescription will vary based on a patients' diagnosis. Diagnoses related to chronic pain conditions have been suggested as an explanation for higher long-term opioid use among primary care patients[19,20], and may partially explain the significantly raised risk seen in this study. The lower risk for long-term use in individuals with a recent external injury is likely due to the more acute nature of pain and shorter follow-up management required of such injuries[1,8]. Overreliance on opioids for the treatment of chronic pain can arise due to a combination of factors including urgency for treatment among this vulnerable group of patients, demonstrated effectiveness for acute pain, and limited alternatives[4]. However, the resultant increases in risk of diversion of prescriptions, overdose, and addiction should be carefully considered by primary health care prescribers[4]. Addiction affects up to one-third of patients with chronic pain[21].

This strictly defined cohort provide insights into and additional support for the risk factors for long-term opioid use. We found that a history of addiction to substances other than opioids and having received treatments for common psychiatric disorders were associated not only with increased risk of opioid use for >3 months, but also with all durations of long-term use investigated. Underlying mental health problems are an important risk factor of opioid use, addiction, and overdose[10,22]. These results are in line with previous findings that showed greater rates of long-term opioid use or addiction among individuals with underlying psychiatric disorders and psychosocial problems[23]. Assessment of such patient-level risk factors by prescribers are recommended in prevention efforts[4].

Among the sociodemographic characteristics investigated, lower levels of education and not cohabiting were also significantly associated with all durations of long-term use. We also saw an association between older age groups and longer durations of use in both models, even though the population was limited to adults under the age of 65. Some previous studies have shown mixed results regarding age and long-term opioid use[9,10], but this study provides additional support for an increased risk among older users. This study also provides evidence of a possible association between living in smaller towns/rural areas or medium-sized towns, as well as being born outside Sweden, and longer durations of long-term use of opioids. Although the results require future investigation, a possible explanation may

be that municipality of residence and country of birth are proxies for socioeconomic status, which may be related to prevalence of multimorbidity and pain conditions, which in turn are associated with long-term use. Although these characteristics may offer additional insight into risk, it should be noted that the most important risk factor for developing addiction is not a trait of the individual patient, but rather the receipt of an opioid prescription itself, due to the pharmacological properties of opioids and their potential for tolerance and dependence[3,4]. Prescribers have a responsibility to carefully weigh the risks and benefits of opioid treatment on a case-by-case basis, while being mindful of dose and duration for all patients[3].

In this cohort, an increasing proportion of individuals receiving an initial prescription for one of several strong opioids—buprenorphine, tapentadol, or fentanyl—progressed to longer durations of use. Long-acting formulations of these drugs may particularly increase the risk of prolonged use[22,24–28]. We did not observe higher proportions of long-term use among individuals initiating on other common strong opioids: morphine and oxycodone. This surprising result may be due to the largely consistent proportion of individuals initiating with short- and long-acting formulations of these opioids across different durations of use, as well as differences in prescribing trends and formulation choices in Scandinavia compared to previous study contexts. We did, however, observe that a higher proportion of individuals with increasing durations of long-term use were initially prescribed tramadol, which was likely the driver of the finding that individuals initiating on weak opioids had a higher proportion of long-term use, since use of codeine combination products remained stable during the study period. This is an important finding since among weak opioids, tramadol is strongly associated with long-term use[24,25,27,28]. We also found that a higher proportion of individuals with long-term use received higher total MMEs and more fills of opioids at initiation compared to short-term users, which is in line with previous research[23–27,29,30].

A strength of this study was the use of comprehensive register data, which enabled us to assess important risk factors[10] and estimate long-term use in the total population of Sweden. Sweden has a publicly financed healthcare system with reimbursement for opioid prescriptions from both public and private providers, which results in near full coverage in the National Prescribed Drug Register. We used data on package-size of pharmacy-dispensed prescriptions to measure opioid use, as we did not have individually written dosing instructions from the prescriber to the patient. We were able to include all dispensed opioids, irrespective of route of administration or strength, and present total opioid use in MMEs, which facilitates international comparison. One caveat is that individuals with unspecified opioid formulations or dose strength were excluded from the MME-specific calculations. However, this was a small proportion, 5.4% (n = 40,622), of the cohort.

Several aspects of the design contributed to the robustness of the study. The strict upper exclusion on age (<65 years) was included to minimize the risk of unmeasured comorbidity among older persons who generally have a higher burden of disease and conditions that could require opioid treatment. The criteria of a ≥ 5-year opioid-naïve period is long compared to most studies, which usually require only 6 or 12 opioid-free months before inclusion[9]. By including a longer opioid-naïve period, we were able to decrease the risk of prior opioid use in the cohort, which can be associated with subsequent use. We assessed risk of opioid use lasting ≥3 months—the most commonly used definition of long-term use[10,11]—as well as the risk associated with longer durations of use. While our use of a 180-day refill gap aligns with previous research[10,13], definitions and modeling approaches vary considerably across studies, leading to differences prevalence estimates[9]. We chose to define durations of opioid use based on the time intervals between prescription refills, without incorporating information on days' supply, due to variability in dose and indication across the substances and formulations assessed, which would require further assumptions in the model. However, the risk of misclassifying more individuals as long-term users in the primary analysis was mitigated by assessing multiple durations of opioid use and conducting sensitivity analyses with stricter refill criteria, which offer additional insight into cases of sustained, near continuous opioid use. This design enhances the robustness of our findings and facilitates meaningful comparisons with other studies.

The study includes key health care-related variables, including history of cancer, which is an indication for opioid treatment that is known to lead to a higher risk of long-term use[1,31]. However, treatment for cancer-related pain does not explain the trends of long-term opioid use seen in the remainder of the cohort, as this group comprised only 3.2%. There were several predictors of long-term opioid use that were not available in the data, including information on somatic diagnoses related to chronic pain from primary care[24,27]. It was therefore not possible to systematically assess the indication for prescription in primary care. Additionally, we had no data on medications administered in inpatient care (e.g., hospitals, palliative care) nor illegal opioid use, which could have led to misclassifying a small number of individuals as opioid naïve.

In summary, 8.1% of this national cohort developed long-term prescription opioid use following an initial opioid prescription. Individuals with psychiatric history, including previous substance use disorders, and certain sociodemographic characteristics were at higher risk of prolonged opioid use. These findings highlight the importance of thoroughly evaluating patients' health status before initiating opioid treatment and ensuring regular follow-up for those prescribed opioids. Additionally, since individuals who received their first prescription in primary care had higher odds of prolonged use, targeted interventions in primary care settings may help reduce long-term opioid use and its associated risks.

## Data availability
Researchers can get access to data for all variables in the dataset by request to National Board of Health and Welfare and Statistics Sweden.

## Code availability
The code used to generate the results in this study is available at Zenodo [31] under https://doi.org/10.5281/zenodo.17038451.

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

## Acknowledgements

This research was supported by the Swedish Research Council for Health, Working life and Welfare (FORTE grants 2024-0267 and 2020-00467) and by research grants from Region Stockholm (ALF). The funders had no role in the analysis or decision to submit this manuscript for publication. We would like to thank scientific editor Kimberly Kane for useful comments on the text.

## Author contributions

C.K., M.G., J.F., and J.W. conceptualized and designed the analysis; C.K., J.H., and H.W. did the data analysis and created the tables and figures. C.K. wrote the first draft of the manuscript. All authors interpreted the results and critically revised and approved the final version of the manuscript. All listed authors meet authorship criteria.

## Funding

## Competing interests

The authors declare no competing interests.
