## [Transparent Peer Review file · Communications Medicine]

A national cohort study of long-term opioid prescription and sociodemographic and health care-related risk factors

Corresponding Author: Ms Cecilia Krüger

Version 0:

Reviewer comments:

Reviewer #1

(Remarks to the Author)

This study is well-conducted and clearly written. It examined long-term opioid use and associated risk factors in a five-year opioid-naïve cohort in Sweden. My primary concern lies in the definition of long-term opioid use, which the authors define as >3 months of opioid use, allowing a refill gap of up to 180 days. Using this definition alone—without incorporating the number of prescriptions or total days' supply—can result in misclassification. For example, a patient who receives two opioid prescriptions 180 days apart could be categorized as a long-term user, which is unlikely to reflect sustained or clinically relevant opioid use. Long-term use typically implies continuous or near-continuous use, which is particularly important to distinguish in the context of opioids, where short-term and as-needed prescriptions are common.

While the authors cite two references to support their definition, I note that Reference 11 includes an additional criterion of ≥ 90 dispensed administration units (e.g., tablets), which adds necessary specificity. Therefore, the alternative definition based on a ≥ 30 -day supply may better align with clinical relevance in the field.

A minor suggestion: when describing the data source, it would be helpful to briefly introduce the included population and which opioid medications are available in Sweden, for readers who may not be familiar with the country's healthcare context.

Another minor point: the manuscript refers to "individuals treated in specialized care for a recent external injury." However, "external injury" and "specialized care" are two distinct variables. It would be clearer to separate these terms to avoid potential confusion.

Reviewer #2

(Remarks to the Author)

This is a very well written manuscript on opioid prescribing and subsequent long-term opioid use among opioid naïve adults in Sweden. The study used registry data to look at short- and long-term opioid use, and risk factors associated with long-term use. The authors found that being older, having lower education, and living alone were associated with long term use. They also found that those with a history of a substance use disorder had higher odds of long-term use.

The rationale for this study and the methodology is clearly described and justified. The statistical methods appear appropriate. The findings appear novel for the Swedish context. A main strength of the study is the robust data made available from the Swedish registries. The references appear comprehensive and up to date.

My first main suggestion would be to improve the clarity of the aims. Since this article includes several characteristics (sociodemographic, opioid, and health-related), clarifying the aims may help the readability. An additional suggestion would be to more deeply consider for the discussion the implications the findings have for prescribers. Also, the patients themselves are not necessarily the risk factors, but inappropriate or over-prescribing of addictive drugs contributes towards the propensity for long-term use (see Dowell, D et al.s' "Opioid analgesics- risky drugs, not risky patients"). In my opinion, the findings from this study could warrant stronger language on the role of primary care prescribers in preventing this public health issue.

Minor comments:

1. Abstract: This is clear and nicely written, I have no comments.
2. Plain language summary: Consider simplifying some of the language. For example: "This study examined the risk of long-term opioid use among adults in Sweden who have never used opioids." Also consider giving an example of what 'external injuries' are.
3. Introduction: In the first paragraph, the authors list some of the negative outcomes associated with prolonged opioid use. I would suggest the authors also include overdose (both non-fatal and fatal), as they are the most serious negative outcome.
4. Introduction: The aims may need clarifying: as it reads, "The aim of this study was to examine long-term opioid use and identify risk factors.." for ? Examine risk factors associated with long-term use? Explore characteristics of patients that developed long-term use? I think some clarification would help the readability.
5. Introduction: In the last paragraph, I suggest removing the last sentence that includes the implications of the study.
6. Conclusion: Consider revising to improve readability. Particularly this sentence: "The findings that individuals with history of addiction, psychiatric disorders, and certain sociodemographic characteristics have a higher risk for continuing to use opioids over longer durations indicates the importance of enhanced evaluation of patients' health status at initiation and the need for regular follow-ups among individuals prescribed opioids."

Thank you for the opportunity to review this important and well-written manuscript.

Sincerely,
Desiree Eide

Version 1:

Reviewer comments:

Reviewer #1

(Remarks to the Author)

The authors did an excellent job addressing all my comments.

Reviewer #2

(Remarks to the Author)

The authors have responded to all of my comments. I have no further comments.

Dear Referees,

Thank you for taking the time to review our manuscript, "*Sociodemographic and health care-related characteristics and risk of long-term opioid prescription: a national cohort study*" with ID COMMSMED-25-0770A.

Your comments were closely considered, and changes to the manuscript were made in accordance with your suggestions. Please see our point-by-point responses to your comments below, with reference to page and line numbers where changes were made. We have also highlighted these lines in the uploaded manuscript file to aid your review.

We hope you feel as we do, that the changes have improved the overall quality of the manuscript. Thank you.

Sincerely,

Cecilia Krüger, MMSc
on behalf of all the authors

Referee expertise:

Referee #1: opioid prescriptions, cohort studies, sociodemographics

Referee #2: opioids, cohort studies

Reviewers' comments:

Reviewer #1 (Remarks to the Author):

1. This study is well-conducted and clearly written. It examined long-term opioid use and associated risk factors in a five-year opioid-naïve cohort in Sweden. My primary concern lies in the definition of long-term opioid use, which the authors define as >3 months of opioid use, allowing a refill gap of up to 180 days. Using this definition alone—without incorporating the number of prescriptions or total days' supply—can result in misclassification. For example, a patient who receives two opioid prescriptions 180 days apart could be categorized as a long-term user, which is unlikely to reflect sustained or clinically relevant opioid use. Long-term use typically implies continuous or near-continuous use, which is particularly important to distinguish in the context of opioids, where short-term and as-needed prescriptions are common.

While the authors cite two references to support their definition, I note that Reference 11 includes an additional criterion of ≥ 90 dispensed administration units (e.g., tablets), which adds necessary specificity. Therefore, the alternative definition based on a ≥ 30 -day supply may better align with clinical relevance in the field.

Thank you for your thoughtful and constructive feedback. We appreciate your recognition of the study's strengths and your concern regarding the definition of long-term opioid use.

We fully acknowledge the complexity of defining long-term use and agree that different modeling approaches can yield varying estimates. In planning our analysis, we considered several alternative definitions, including those incorporating days' supply or number of dispensed units, such as the approach used by Hamina et al. [Reference 11]. However, in a nationwide study encompassing all opioid types and routes of administration, we found that incorporating days' supply would require substantial assumptions about dosing regimens and formulation-specific administration (e.g., tablets, patches, liquids). Opioid dosing also varies widely by indication, and since we included all prescriptions regardless of indication, we felt

these assumptions could introduce additional misclassification—particularly given the absence of dose-specific information in our dataset.

Given these limitations, we selected a refill-gap approach that allows for inclusion of all opioid formulations and minimizes assumptions about dosing. While we recognize that a 180-day refill gap is relatively generous and may classify some individuals with sporadic prescriptions as long-term users, this definition gives a good overview of prescribing durations on a national level, is consistent with prior literature, and allows for international comparability.

To address the limitations of this approach, we conducted a sensitivity analysis using stricter classifications of use to see how associations with risk factors may change with a more narrowly defined population of long-term users. In the analysis presented in the previous draft (Supplementary Figure 1, Supplementary Table 3; Results pages 15-16) we shortened the refill gap to 120 days, which reduced the proportion of individuals classified as long-term users by 3%. Importantly, which risk factors were associated with long-term use largely remained the same and the magnitude of the estimates increased, supporting the robustness of our primary findings.

In response to your comment, we conducted an additional sensitivity analysis using an even more stringent definition: a 90-day refill gap. In practice, this required individuals to have at least two prescriptions within days 0–90 and at least one additional prescription during days 91–180. This definition more closely approximates sustained, near-continuous opioid use. The results were directionally consistent with our sensitivity analysis with the 120-day gap, and showed even higher estimates, further supporting the validity of our findings.

We have updated the manuscript to reflect this new sensitivity analysis (Methods: page 9, lines 164–69; Results: pages 15-16, lines 275–297; Supplementary Figure 2 and Supplementary Table 4). In the updated Results section, we discuss the two sensitivity analyses together, highlighting how stricter criteria influence the estimates. In the limitations section of the Discussion, we now explicitly include the rationale for the selected refill-gap approach while acknowledging that it may increase the risk of misclassification (page 20, lines 377–87). We also clarify that inclusion of multiple definitions and sensitivity analyses strengthens the study's design and enhances comparability with other research.

We hope this additional analysis and clarification addresses your concern and demonstrates our commitment to methodological transparency and rigor.

2. A minor suggestion: when describing the data source, it would be helpful to briefly introduce the included population and which opioid medications are available in Sweden, for readers who may not be familiar with the country's healthcare context.

We agree that this would be a valuable addition to the article, as it would set the scene more clearly for international readers. We have added a description of which

medications are available and commonly used in Sweden in section on *Opioid use and duration of use* on pages 6-7, lines 98-109. In line with this change and so as not to be repetitive, we removed the list of substances from the section on *Opioid characteristics* on page 8.

3. Another minor point: the manuscript refers to “individuals treated in specialized care for a recent external injury.” However, “external injury” and “specialized care” are two distinct variables. It would be clearer to separate these terms to avoid potential confusion.

Thank you for this helpful observation. To avoid confusion, we have removed references to “specialized care” when discussing the external injury variable throughout the manuscript (Abstract Page 2, lines 12-14 and 18-19; page 17, lines 303-04; page 18, line 316). We have retained the mention of specialized care only in the section on *Health care-related characteristics*, where we define the variable (page 8, lines 137-38).

Reviewer #2 (Remarks to the Author):

1. This is a very well written manuscript on opioid prescribing and subsequent long-term opioid use among opioid naïve adults in Sweden. The study used registry data to look at short- and long-term opioid use, and risk factors associated with long-term use. The authors found that being older, having lower education, and living alone were associated with long term use. They also found that those with a history of a substance use disorder had higher odds of long-term use.

The rationale for this study and the methodology is clearly described and justified. The statistical methods appear appropriate. The findings appear novel for the Swedish context. A main strength of the study is the robust data made available from the Swedish registries. The references appear comprehensive and up to date.

Thank you your thoughtful and encouraging comments. We appreciate your positive feedback on the manuscript.

2. My first main suggestion would be to improve the clarity of the aims. Since this article includes several characteristics (sociodemographic, opioid, and health-related), clarifying the aims may help the readability.

Thank you for your suggestion that we improve the clarity and readability of the study aims. We have revised the aim to read: "The aim of the study was to analyze the risk of long-term use after an initial opioid prescription in a national cohort of five-year opioid-naïve adults and examine associated sociodemographic and health care-related risk factors." (changes made on Page 2, Abstract, lines 3-5; Page 4, lines 56-8).

3. An additional suggestion would be to more deeply consider for the discussion the implications the findings have for prescribers. Also, the patients themselves are not necessarily the risk factors, but inappropriate or over-prescribing of addictive drugs contributes towards the propensity for long-term use (see Dowell, D et al.s' "Opioid analgesics- risky drugs, not risky patients'). In my opinion, the findings from this study could warrant stronger language on the role of primary care prescribers in preventing this public health issue.

Thank you for this thoughtful suggestion. We agree that potentially inappropriate prescribing plays a critical role in the development of long-term opioid use, and we appreciate the reference to Dowell et al., which we have now cited in the revised Discussion with commentary (pages 18-19, lines 340-45). Our intention was not to imply that patients themselves are the risk factors, but rather to describe associations observed in the data. As our study did not include information on the clinical indication for opioid prescriptions, we were cautious about making direct claims regarding prescribing appropriateness. Nonetheless, we recognize the

importance of prescriber behavior in this context and have strengthened the language regarding the role of primary care prescribers in the Discussion as well (see page 18, lines 317-21, 328-29).

Minor comments:

1. Abstract: This is clear and nicely written, I have no comments.

Thank you.

2. Plain language summary: Consider simplifying some of the language. For example: “This study examined the risk of long-term opioid use among adults in Sweden who have never used opioids.” Also consider giving an example of what ‘external injuries’ are.

Thank you for the helpful suggestion regarding the plain language summary. We have incorporated edits in line with your proposal and made additional changes to further simplify the wording, aiming to make the results more accessible to a broader audience. We also added an example to clarify what we mean by ‘external injuries’. Please see the updates on page 3, lines 25–35.

3. Introduction: In the first paragraph, the authors list some of the negative outcomes associated with prolonged opioid use. I would suggest the authors also include overdose (both non-fatal and fatal), as they are the most serious negative outcome.

Thank you for this important suggestion. We agree that overdose is among the most serious potential consequences of prolonged opioid use. We have revised the Introduction to include this point explicitly (page 4, lines 38-41).

4. Introduction: The aims may need clarifying: as it reads, “The aim of this study was to examine long-term opioid use and identify risk factors..” for ? Examine risk factors associated with long-term use? Explore characteristics of patients that developed long-term use? I think some clarification would help the readability.

Please see our reply above to Major Comment 2 above. We hope this clarification improves readability.

5. Introduction: In the last paragraph, I suggest removing the last sentence that includes the implications of the study.

We acknowledge that including the implications of the study in the Introduction may be somewhat unconventional. However, we included this sentence (page 4, lines 60-62) in accordance with the *Communications Medicine* Style and Formatting Checklist for Primary Research Articles, which recommends that the final paragraph

of the Introduction provide a summary of the major results and conclusions. That said, we are happy to revise or remove this sentence should the editor prefer a different structure.

6. Conclusion: Consider revising to improve readability. Particularly this sentence: “The findings that individuals with history of addiction, psychiatric disorders, and certain sociodemographic characteristics have a higher risk for continuing to use opioids over longer durations indicates the importance of enhanced evaluation of patients' health status at initiation and the need for regular follow-ups among individuals prescribed opioids.”

Thank you for the opportunity to review this important and well-written manuscript.

Thank you for your helpful suggestion. We have revised the conclusion to improve clarity and readability. In particular, we rephrased the sentence you highlighted to better convey the key findings and their implications in a more concise and accessible manner. We also made minor edits throughout the paragraph to enhance flow and precision (pages 20-21, lines 398-405).

We are grateful for your insightful comments, which have significantly contributed to improving the manuscript. Thank you.